# Attachment of Fibrinogen on Ion Beam Treated Polyurethane

**DOI:** 10.3390/biomimetics9040234

**Published:** 2024-04-15

**Authors:** Vyacheslav Chudinov, Igor Shardakov, Irina Kondyurina, Alexey Kondyurin

**Affiliations:** 1Institute of Continuous Media Mechanics, Ural Branch, Russian Academy of Sciences, Perm 614013, Russia; chudinovsl@mail.ru (V.C.); shardakov@icmm.ru (I.S.); 2School of Medicine, University of Sydney, Camperdown, NSW 2050, Australia; i.kondyurina@gmail.com; 3Ewingar Scientific, Ewingar, NSW 2469, Australia; 4School of Physics, University of Sydney, Camperdown, NSW 2050, Australia

**Keywords:** protein, covalent attachment, polyurethane, ion beam treatment

## Abstract

Protein-stable coverage of the artificial implant is a key problem for biocompatibility. In the present study, a protein layer was attached covalently to a polyurethane surface treated by an ion beam. A plasma system consisting of a vacuum chamber (0.8 Pa pressure) with a high voltage electrode powered by a short pulse (20 μS pulse duration and 200 Hz pulse repetition) generator was designed. Polyurethane with a formulation certified as a material for medical implants was treated by nitrogen ions with an energy of 20 keV and 5 × 10^14^–10^16^ ions/cm^2^ fluence range. Wettability measurements, X-ray photoelectron, Raman, Fourier transform infrared attenuated total reflection, and ellipsometry spectra showed a significant change in the structure of the surface layer of the treated polyurethane. The surface of the treated polyurethane contained a carbonised layer containing condensed aromatic clusters with terminal free radicals. The surface energy of polyurethane surface increased from 33 to 65 mJ/m^2^. The treated polyurethane surface became capable of adsorbing and chemically binding protein (fibrinogen). The designed system for ion beam treatment can be used for surface activation of biomedical polymer devices, where a total protein coverage is required.

## 1. Introduction

The nonspecific interaction of the surface of a biomedical device with molecules from an organism or a biotechnological process is a constant obstacle to the use of artificial materials in biomedical technologies [1,2,3]. This interaction leads to a change in the conformation of biologically significant molecules and their activity, disrupting their bio functionality [4,5]. The best way out of this problem is to modify the surface of an artificial material by applying a permanently bound total layer of biological molecules on the surface while preserving their conformation and activity [6,7,8,9].

In some cases, it can be achieved by using precursor intermediate molecules, with one side attached to the surface of the device and the other providing a covalent attachment of the protein to the surface [10,11,12,13,14]. There are a number of publications where the surface of a polymer implant was treated for chemical binding with protein. For example, some applications involving protein attachment have achieved success by utilising linkers such as cysteamine [15], N-hydroxysuccinimide, N-(3-dimethylaminopropyl)-N′- ethyl carbodiimide hydrochloride [16,17], glutaraldehyde [18,19,20,21,22,23], succinimidyl 3-(2-pyridyldithio)propionate and 2-pyridylthio benzophenone PDT-BzPh photolinker [24]. These surface preparation methods are convenient for protein attachment. However, toxic reagents have to be used, which is undesirable for biomedical implants of organisms. In addition to that, the covalently attached protein covers only a part of the implant surface, but complete surface coverage is required. Therefore, in some cases, the use of these precursors is unacceptable.

As was shown in some cases, plasma activation of the surface and direct covalent grafting of the protein and other molecules can be used without any precursors. For this purpose, ion beam treatment methods were used for polymer implants, plasma immersion ion implantation (PIII), and plasma polymerisation methods. These are very powerful methods that cause carbonisation, oxidation, and depolymerisation of the thin surface layer of different polymers while preserving the bulk underneath layers [25,26,27,28,29,30,31,32,33,34,35,36], including polyurethanes (PUs) [32,33,34,35,36,37,38,39,40,41,42]. Some examples of these systems are the ion beam implanter ILU [43], the ion beam implanter Pulsar at the Institute of Technical Chemistry in Perm [44,45], PIII systems at Rossendorf Research Center in Dresden [46] and Leibniz Institute of Surface Engineering in Leipzig [47], and PIII and plasma polymerisation systems at the University of Sydney [47].

The complete coverage of the surface with protein has been demonstrated, in particular, during plasma immersion implantation of a number of polymers. Proteins have been covalently attached to polyethylene (PE), polystyrene, polytetrafluorethylene, Nylon, polyvinyl chloride, polypyrrole, polycarbonate, polyether ether ketone, polyether sulfone, PUs, polycaprolactone, and polydimethyl siloxane without the use of a precursor. The following have been demonstrated to be covalently attached and maintain catalytic activity: poly-L-lysine and Catalase, Horseradish peroxidase, β-glucosidase, tropoelastin, collagen, fibronectin, poly-L-glycine, Poly-DL-alanine, Poly-L-isoleucine, poly-L-lysine, Poly-L-histidine, Poly-L-tryptophan, Poly-L-tyrosine, Poly-L-Threonine, Poly-L-methionine, osteocalcin, and antibodies CD2, CD3, CD29, CD184, CD244.1, CD326, and CD34. Additionally, covalently attached oligonucleotides have also been shown. In this case, the protein coating was applied to 92% of the polymer surface area. The remaining area of the surface (without the protein) was not accessible to the protein molecules due to steric hindrance caused by neighbouring molecules. Therefore, the polymer surface area can be considered fully covered (100%). Protein grafting onto the treated polymer surface is based on the reaction of free radicals located at the edges of graphite clusters on the polymer surface. The universality of free radical chemical activity allows the protein molecule to graft via any amino acid residue [48].

An example of successful protein grafting is the result of eliminating a foreign body reaction in an organism by implanting a PIII-activated PU implant in an animal. This implant shows a significantly weakened rejection reaction and, in some cases, the absence of a foreign body reaction [49]. The grafted protein hides the surface of the PU implant from the immune cells, and the organism does not recognise the implant as a foreign body. An analysis of the grafted proteins on the implant surface extracted from the animal showed the presence of a number of important functional proteins, such as collagen, fibrinogen, and others [50,51,52,53,54]. In the present study, we investigate the chemical attachment of fibrinogen to the surface of medical PU for implants.

For surface activation, the polymer must be treated with a high-energy ion beam. For this purpose, ion accelerators can be used. The polymer is placed in a vacuum chamber, and a high vacuum is created. Then, a plasma is formed separately, from which a beam of ions is extracted and accelerated. The ion beam can be separated from other particles by composition and energy. An ion accelerator is a fairly large and expensive machine. The area of the beam spot is about 1 cm^2^. There are also accelerators without ion beam separation with a larger aperture, which allow processing polymer areas up to 100 cm^2^ [55]. The ion beam in these systems is unidirectional, which limits the shape of the polymer being processed. The strong vacuum requirement in these systems limits the class of polymers processed, which must have a low rate of outgassing.

A more technologically advanced method is PIII [47,56]. The PIII technique involves placing a high-voltage electrode with a polymer in a vacuum chamber. The polymer is covered with a metal grid connected to a high-voltage electrode to exclude a changing effect [47]. The radio frequency or microwave plasma is generated in the chamber. A short high-voltage pulse is applied to the high-voltage electrode, which pulls ions out of the plasma and accelerates them in a direction perpendicular to the grid. The ions are accelerated through the grid and bombard the polymer. In this case, the grid and the processed polymer can be of a complex shape [57]. This method is more suitable for processing real polymer implants used in medicine and biomedical technologies.

However, when PIII plasma is used in a vacuum chamber, the plasma cloud usually touches the walls of the chamber. As a result of plasma bombardment of the chamber walls, the wall material can be etched, and the chamber gas environment can be contaminated. The authors had these negative experiences. As a result, the products of etching of the chamber walls can approach the polymer surface being processed and deactivate it. In this case, the PIII could not activate the polymer surface, and chemical grafting of the protein could not be conducted. To eliminate this effect, it is desirable that the plasma in the chamber does not touch the chamber walls or other parts of the system.

As an option to eliminate the contact between the plasma and the chamber wall, we have designed a different ion beam implanter based on a different principle: no additional plasma is generated in the chamber, and a high-voltage pulse is applied only to the high-voltage electrode. Then, during a high-voltage pulse, plasma appears above the surface of the electrode, and at the same time, the ions are pulled out and accelerated toward the high-voltage electrode. With a sufficiently large space above the electrode, the plasma sheath does not reach the chamber walls, and the influence of the etching of the wall material is eliminated. The size of the electrode and, consequently, the area of the treated polymer is limited only by the volume of the vacuum chamber. The shape of electrodes and polymer implants can be complex. In addition, this eliminates the need to generate radio frequency or microwave plasma in the chamber, which simplifies the system design and makes its operation more stable. This new ion beam treatment method was tested in the present study.

## 2. Materials and Methods

In the experiments, polyurethane SKU-PFL, which is certified for medical use and long-term implants in the human body, was used [58,59,60]. PU was synthesised from polytetrahydrofuran with terminal hydroxyl groups terminated by toluene diisocyanate. MOCA aromatic diamine was used for curing. The formula of PU is shown in Figure 1. PU films 0.3 mm thick were prepared for this study. Before treatment, PU films were swollen in toluene to wash out unreacted products, and then the films were dried until the solvent was completely removed. The presence of solvent and unreacted products was monitored by FTIR ATR spectra and mass loss measurements.

The PU samples were treated in a plasma system consisting of a vacuum chamber, turbomolecular pump, scroll dry pump, and high-voltage pulse generator (Figure 1). The pressure was measured with a digital vacuum meter. The ultimate pressure of the residual atmosphere was down to 0.013 Pa (0.1 mTorr). The working pressure of nitrogen gas was 0.8 Pa (6 mTorr). The samples were placed on a high-voltage electrode and covered by a metal grid. High voltage pulses of −20 kV were applied. The pulse duration was 20 μs, and the pulse frequency was 200 Hz. The pulse voltage and current were measured with digital oscilloscopes. The ion fluence was calibrated using UV spectra of the treated PE film following the method described in [47]. UV transmission spectra of PE films for the fluence calibration were recorded on a Varian spectrophotometer.

The treated PU samples were incubated in fibrinogen solution (20 g/L in a 10 mM sodium phosphate buffer, pH = 7) overnight at room temperature. Each sample was treated in an individual tube. After that, the samples were washed 6 times in a new tube and fresh buffer each time. Each wash in buffer was 10 min. After that, the samples were rinsed for a few seconds in mQ water to remove the buffer salts and were then dried overnight. For control, the untreated PU samples were incubated and washed under similar conditions. Other control-treated and untreated PU samples without protein were incubated only in buffer solution and rinsed in mQ water under similar conditions at the same time as the samples with protein. Then, the FTIR ATR spectra of all these samples were recorded.

After the spectra recording, all treated and untreated samples with and without protein were washed in sodium dodecyl sulphate (SDS) detergent solution of 2% at 70 °C for 2 h to remove all physically adsorbed protein. Each sample was washed in an individual tube. After that, the samples were washed in mQ water 3 times, each time in a new tube to remove the SDS detergent. Each sample was washed in an individual tube and then dried overnight. The FTIR ATR spectra of dried samples were recorded.

The samples were measured with the FTIR (Fourier transform infrared) spectrometer Digilab (Digilab, Hopkinton, MA, USA) equipped with an ATR (attenuated total reflection) accessory. The prism crystal of Ge with a 45-degree incident angle and 25 reflections was used as the ATR element. The 500 scans for each spectrum with a spectral resolution of 4 cm^−1^ were used. The surface layer of the samples was measured with a Woollam M2000V spectroscopic ellipsometer. The measurements were conducted for three angles, 65, 70, and 75 degrees, in a range of 200–1000 nm wavelength. The Cauchi layer was used for the optical model. The KRUSS contact angle analyser was used for water and diiodomethane contact angle measurements. The optical images of the sample surface were obtained using a Carl Zeiss Jena microscope with a digital camera. X-ray photoelectron (XPS) spectra were recorded using a SPECS spectrometer. Additionally, an Al K_a_ monochromatic X-ray source was used. Survey spectra in an eV region of 50–1400 eV and high-resolution spectra of C_1s_, N_1s,_ and O_1s_ regions were recorded. The SRIM-2003 code was used to calculate ion penetration depth into the PU [61,62].

## 3. Results

### 3.1. Evaluation of Treatment Parameters

The choice of treatment regime was based on previously obtained results on the activation of the polymer surface during ion beam treatment in the Pulsar implanter and PIII systems. The treatment parameters of the new system were calibrated to obtain the same result as the PE film treatment. The maximum voltage of 20 kV was determined by previous successful polymer processing results. The shape of the high-voltage pulse was close to rectangular, which ensured, if possible, that only ions of a given energy were processed. Figure 2 shows diagrams of the voltage and current of the electrode at different pressures in the vacuum chamber. At ultimate pressure (0.1 mTorr) in the vacuum chamber, there was no plasma and a slow voltage drop after the end of the pulse was ensured due to the low discharge current in the electronic circuit of the pulse source. At pressure lower than 5.5 mTorr, no plasma current and slow discharge current were observed in the chamber. As the pressure increased, the discharge current pulse appeared at the end of the pulse. In this case, the voltage drop after the pulse occurred due to a gas discharge through the plasma formed during the pulse. When the pressure increased to 7.3 mTorr, the discharge current reached the maximum values provided by the high-voltage source. The shape of the voltage pulse became rectangular, showing a decrease in values at the end of the pulse. These measurements made it possible to select the working gas pressure with maximum processing efficiency and stable ion energy set by the voltage on the high-voltage electrode. Note that the main part of the ion current in this mode occurs at the end of each high-voltage pulse, ensuring the accelerating voltage’s stability for all ions throughout the entire pulse.

The pulse repetition frequency of 200 Hz was limited by the requirement that the polymer samples should not overheat. With an increase in the pulse repetition frequency, the PE film used as a control was deformed. This was the boundary of the non-thermal treatment. At 200 Hz, the arrival of the next pulse occurred after 5 mS, which is much longer than the relaxation time of the discharge current and voltage after the passage of the first pulse observed in the diagram. Further studies were carried out with the following selected optimal parameters: a gas pressure of 6 mTorr, a pulse repetition frequency of 200 Hz, a pulse duration of 20 μS, and a voltage amplitude of 20 kV.

A calibration of the ion fluence based on previous results was performed using the UV absorption spectra of the PE film. For this purpose, PE films 20 μm thick were treated with nitrogen ions of 20 keV energy, as it was conducted previously with ion beam treatment and with PIII of PE [47]. The UV spectra of PE and absorption at selected wavelengths as a function of treatment time are shown in Figure 2.

The UV absorption spectra of the treated PE samples showed increasing absorption with treatment time, similar to the previously observed absorption of PE films after ion beam treatment and PIII. The absorption in the short-wavelength region of the spectrum increases faster than in the long-wavelength region of the spectrum. The dependence of absorption on treatment time shows that an increase in light absorption occurs along an asymptotic curve with saturation. Saturated values of absorption correspond to the maximum degree of carbonisation of the surface layer of the PE, with the thickness corresponding to the penetration depth of the bombarding ions.

The results of the absorption in PE films, depending on the treatment time, were matched with the calibration curve obtained by treating the same PE films with nitrogen ions of 20 keV energy in an ion implanter with a known treatment fluence. This made it possible to interpret further results in the new system in terms of fluence, that is, 2 × 10^13^ ions arriving per 1 cm^2^ surface area of the polymer per 1 s of the treatment.

To confirm the carbonisation of the PE surface in this treatment, micro-Raman spectra of the surface layer of PE were recorded (Figure 3). The spectrum shows the presence of D-peak at 1355 cm^−1^ and G-peak at 1534 cm^−1^ of the carbonised structure. According to the Raman model of the carbon spectrum [63], the ratio of peak intensities and their positions correspond to the nc-graphite phase with the amorphous carbon phase. The L_a_ cluster diameter is about 5 nm. A similar structure in the surface layer was observed after PIII and ion beam treatment of PE.

Thus, the present plasma system creates a carbonised layer on the surface of PE films similar to the carbonised layer resulting from PIII and ion beam treatment in ion implanters. This plasma system and optimal parameters for the non-thermal regime of treatment were used to activate the surface of medical PU for protein adsorption.

### 3.2. Ion Beam Treatment of Polyurethane

The surface of the PU film is smooth after the synthesis (Figure 4). The morphology of the untreated PU surface is mirrored by the surface profile of the substrate on which the PU film is formed. The treated PU surface is covered with cracks. PU is a highly elastic material and has an elongation at a break of 300% [60]. Therefore, the appearance of cracks on the surface of the treated PU indicates the formation of a brittle layer and internal stresses in it.

The ellipsometry spectra and the subsequent optical model showed that the top surface layer of the treated PU has different optical parameters than the original untreated PU (Figure 5). The spectra of the original PU show some dispersion of the refractive index and extinction coefficient in the region of wavelengths shorter than 300 nm. This dispersion is due to the absorption of light in the aromatic structures and carbonyl groups of the PU. The refractive index values are in the region of about n = 1.5, which is typical for polymer materials, including polyurethane materials.

The refractive index of the surface layer of the treated PU is higher than that of the original PU. The extinction coefficient of this layer also shows increased values compared to the extinction coefficient of untreated PU.

The extinction coefficient spectra of PU show the same character as the UV absorption spectra for the treated PE. An increase in light absorption is observed higher in the short-wavelength region of the spectrum and lower in the long-wavelength region of the spectrum. This dispersion of the refractive index and extinction coefficient cannot be explained by the aromatic structures and carbonyl groups of the PU itself but can be explained by the formation of condensed aromatic rings and unsaturated carbon–carbon structures with various conjugation lengths.

The refractive index and extinction coefficient of the top layer increase with an increase in ion fluence. The increase has a saturation character. The saturated values are achieved for the treatment time above 400 s. A further increase in the treatment time does not increase the saturated values of the refractive index and extinction coefficient. The refractive index value reaches n = 1.8, which is typical for the refractive index of carbon film in the form of graphite or diamond [64,65].

The thickness of the modified top surface layer of PU, according to ellipsometry data, is about 30 nm. The thickness of the layer does not depend on the treatment time. At a treatment time of 40 s, the layer size is measured about 60 nm. However, the optical model of this sample does not fit the experimental curves well. The refractive index and extinction coefficient spectra have deviations, which can be associated with the inhomogeneity of the modified layer at low treatment time. Therefore, the optical model of ellipsometry spectra for these samples, consisting of two homogeneous materials, is not correct. Unfortunately, it was also not possible to obtain a sufficiently accurate optical model with an effective layer for these samples.

XPS spectra of untreated PU showed the presence of carbon (78.1%), oxygen (19.0%), and nitrogen (2.9%) in the ester, urea, and urethane bonds (Figure 6). The relative concentration of these elements, calculated by the integral intensity of the lines, corresponds to the structural formula of PU.

The spectra of the treated PU showed significant differences. The total nitrogen content increased, and the total oxygen content decreased markedly (Figure 7). The total carbon content did not change, but the carbon content in the carbon–carbon bonds increased from 51.5% to 62.4%. The ether groups’ carbon and oxygen content decreased significantly (Figure 7). The carbon and oxygen content in carbonyl groups increased. New lines of nitrogen in unsaturated double (401.5 eV) and triple bonds (404.4 eV) with carbon atoms appeared. The concentration of elements, calculated by the intensity of these lines, no longer corresponded to the structural formula of PU.

FTIR ATR spectra of untreated PU show lines of ester group, methyl groups, urethane, urea, and aromatic groups according to the structural formula of the PU (Figure 8). The spectra of the treated PU differ slightly in the region of 3700–3000 cm^−1^ and 1800–1500 cm^−1^. Small differences in the spectra are due to the fact that the thickness of the modified layer as a result of ion beam treatment (less than 100 nm) is significantly less than the thickness of the analysed layer in the FTIR ATR spectrum with ATR germanium crystal (400–600 nm). For a detailed analysis, differential spectra were obtained, where the spectrum of untreated PU was subtracted from the spectra of the treated PU. In the differential spectra, new lines are clearly visible in the region of vibrations of the hydroxyl group with maximum at 3480 and 3410 cm^−1^, a complex contour in the region of vibrations of the C=O, C=N and C=C groups with maximum at 1703 cm^−1^, 1670 cm^−1^ and a shoulder up to 1600 cm^−1^, and an intense line in the region of C-O vibrations with a maximum at 1080 cm^−1^.

For quantitative measurements of spectral line intensity, the method of internal standard was used (Figure 9). For this purpose, the intensity of the 3410 cm^−1^ new vibration line of the hydroxyl group was normalised to the intensity of the 2859 cm^−1^ line of methyl groups of PU. The intensity of 1703 cm^−1^, 1670 cm^−1,^ and 1613 cm^−1^ new vibration lines of unsaturated groups was normalised to the intensity of line 1534 cm^−1^ vibrations of Amide II of PU. This intensity normalisation made it possible to take into account the quality of the optical contact of the PU sample with the ATR crystal. For all new lines analysed, an asymptotic dependence of the intensity of these lines on the treatment time was observed. After 400 s of treatment, the increase in the intensity of new lines practically stopped within the standard deviation range.

Raman spectra of PU also showed the formation of a carbonised structure in the surface layer of PU after the treatment (Figure 3). However, in contrast to the Raman spectrum of treated PE, the intensity of the new lines of the carbonised structure in the spectrum of treated PU is low. Observation of the lines of the carbonised structure was possible only in the differential Raman spectrum. The D-line is observed at 1378 cm^−1^, and the G-line is observed at 1610 cm^−1^. The position of these lines and the ratio of their intensities indicate the presence of nc-graphite of 2 nm characteristic size in the surface layer of the treated PU.

The ion beam treatment also changed the water wetting of the PU (Table 1). Untreated PU showed a water contact angle of 94 degrees. The contact angle of the treated PU immediately after removing the sample from the vacuum chamber to the air was 37 degrees. But after a month, the contact angle increased to 57 degrees. The contact angle of the non-polar liquid methyl diiodide was 55 degrees for untreated PU, 39 degrees for freshly treated PU, and 50 degrees for treated PU after long-term storage. Thus, the contact angles of the treated PU did not return to the original values for the untreated PU.

The surface energy and its polar and dispersic components, as calculated by the Owens–Wendt method, showed that the untreated PU surface is highly hydrophobic. The total surface energy of 32.6 mJ/m^2^ is mainly formed by a dispersic component of 31.4 mJ/m^2^. The polar component of 1.2 mJ/m^2^ is neglectable. After the ion beam treatment, the surface energy increases to 65.4 mJ/m^2^ due to the most change in the polar component of 25.3 mJ/m^2^. After a month of storage time, the total surface energy drops to 50.6 mJ/m^2^, with the polar component decreasing to 16.4 mJ/m^2^ the most. Thus, the surface energy of ion beam-treated PU remains sufficiently high after a long storage time.

### 3.3. Attachment of Fibrinogen

The FTIR ATR spectra of PU with fibrinogen coating differ very little from the spectra of the same PU samples without fibrinogen. The intensity of fibrinogen vibration lines cannot be visibly high if the fibrinogen layer is a monolayer. To analyse the spectrum of the applied fibrinogen, the differential spectra of PU were obtained. The spectrum of PU without protein was subtracted from the spectrum of PU with attached fibrinogen. In this case, the conditions and time for ion beam treatment of the PU, the time after ion beam treatment until the application of the fibrinogen solution, the conditions for keeping the PU in a buffer solution, the conditions and time for washing unbound protein in buffer, water, and detergent, the conditions and time for drying the samples were identical for the sample with fibrinogen and a sample without protein. Then, the spectrum of the PU vibrations is subtracted completely, allowing the spectrum of the adsorbed fibrinogen to be clearly observed. The differential spectra of PU with attached fibrinogen are shown in Figure 10.

The spectra are shown for samples of untreated PU and for PU treated with different treatment times. The spectral lines of Amide A at 3300 cm^−1^, Amide I at 1650 cm^−1^ and Amide II at 1540 cm^−1^ as the characteristic vibrations of the amide group in the fibrinogen molecule are clearly observed. The intensity of the fibrinogen lines is much lower than the intensity of the vibration lines of the PU itself and the carbonised surface layer. The fibrinogen spectra look the same for all PU samples, untreated and treated with an ion beam: the position of the lines and the ratio of their intensities do not differ. These spectra refer to the fibrinogen physically and chemically adsorbed on the surface of the PU.

Differential spectra of PU after washing in detergent showed the presence of fibrinogen lines for samples treated with an ion beam only (Figure 10). In the spectra of these samples, the lines Amide A at 3300 cm^−1^, Amide I at 1650 cm^−1,^ and Amide II at 1540 cm^−1^ are visible. In the spectrum of untreated PU, no lines are observed in these regions at the noise level of the spectrum.

To quantify the fibrinogen on the PU surface, the peak intensity of all Amide lines was used. The intensity of the lines was normalised to known extinction coefficients for each individual Amide line determined earlier [48]. After the normalisation, the intensities of all Amide lines were averaged. The results are presented in Figure 11. The averaged values are proportional to the relative concentration of the fibrinogen according to the Bouguer–Lambert law. The concentration of fibrinogen attached to the surface of untreated PU is about half of the concentration of fibrinogen attached to the surface of ion beam-treated PU. The concentration of the attached fibrinogen does not depend on the ion beam treatment time.

The fibrinogen was completely washed out from the surface of untreated PU in detergent, which is observed by the absence of the Amide lines in the spectrum. The concentration of fibrinogen remaining on the surface of the ion beam treated PU after washing in detergent is, on average, about 80% of the concentration of initially attached fibrinogen. The remaining fibrinogen concentration after washing does not depend on the ion beam treatment time.

## 4. Discussion

The results of medical PU treatment using a new plasma system showed that, despite the simplification of the system and treatment process, the activation of the PU surface occurs in approximately the same way as in the case of ion beam treatment and PIII. A layer of carbonised structures, such as condensed aromatic clusters, is formed on the surface, in which the carbon atoms at the edges of the clusters have unpaired electrons. This is observed in the Raman and UV absorption spectra. The characteristic sizes of these clusters, in this case, were 2 nm. For comparison, the size of clusters on the surface of PE treated with nitrogen ions with an energy of 20 keV in ion beam treatment was 2 nm. The sizes of clusters formed as a result of PIII treatment on the surface of polystyrene treated with argon ions with an energy of 20 keV are 1 nm and nitrogen ions with an energy of 20 keV 1.8 nm. The sizes of graphite clusters on the surface of Pebax polyamide treated in PIIII with nitrogen ions with an energy of 20 keV were 1.5 nm, and with an energy of 30 keV, they were 1.8 nm [47]. The sizes of graphite clusters on the surface of PU treated in PIIII with nitrogen ions with an energy of 20 keV were 2.5 nm [49]. The size of graphitic clusters is important for the stabilisation of free radicals in the surface layer after ion beam treatment before a protein solution is applied.

The graphitic clusters that appear in the surface layer of PU contain free radicals. The ESR spectra of the treated PU showed the appearance of unpaired electrons at carbon atoms located in the graphite clusters. Similar signals of unpaired electrons in graphite clusters were observed in PU [49], polyether ether ketone, and PE [48] treated by nitrogen ions in PIII. In this case, the signal of unpaired electrons in the EPR spectra was observed after long-term exposure of the treated PU to air at room temperature. This also corresponds to the literature data on the stabilisation of free radicals at the edges of graphite clusters [66,67,68,69].

In the FTIR ATR spectra of treated PU, rather strong lines of hydroxyl groups are observed. Similar lines were observed in the spectra of PU treated with PIII with nitrogen ions with an energy of 20 keV. These changes correspond to the depolymerisation of the deep layers that lie beneath the surface carbonised layer. This process is apparently caused by the migration of free radicals from the surface carbonised layer into the deeper native layers of the PU.

At the same time, it is necessary to note a much lower degree of carbonisation of the surface layer of PU in this treatment method. Under the same treatment conditions (energy and type of ions, ion fluence), a significantly smaller thickness of the carbonised layer (30 nm) is observed in this method compared to PIII, where the thickness of the carbonised layer is about 70 nm. According to SRIM calculations, the penetration depth of nitrogen ions with an energy of 20 keV is about 70 nm. Therefore, the difference in the depth of penetration of nitrogen ions into PU and the thickness of the resulting carbonised layer of PU remains unclear.

The graphitic clusters with free radicals that appeared on the surface provided a covalent attachment of fibrinogen molecules to the surface of the treated PU. Washing the non-covalently grafted fibrinogen from the surface with a detergent showed that the fibrinogen from the surface of the untreated PU is washed off almost completely, while about 80% of the initially adsorbed fibrinogen remains on the treated PU.

Similar adsorbed protein retention results were observed on other polymers treated with PIII [47,48]. It was shown that after treatment, a continuous monolayer of protein is formed on the surface of the treated polymers. Drawing parallels with previous results, it can be assumed that this treatment also provides a protein coating on the total surface of the treated PU.

Thus, we can assume that this type of treatment can be used to activate the surface of PU for biomedical applications such as non-rejected implants [49], microarray for analysis, improved cell adhesion devices [46], and other biomedical devices where the covalent adsorption of protein onto the surface is a key factor. The precursors for protein covalent attachment are not required. The plasma system has a simple design and can be directly installed in a biomedical laboratory or surgery room. Following studies with animal trials are on the way.

## 5. Conclusions

A simple and efficient system has been created for ion beam treatment of the polymer surface. The system consists of a vacuum chamber with a high-voltage electrode and a pulse generator. This system does not require additional plasma generation in the chamber to produce the ion beam. These studies have shown that the biomedical polyurethane treated by an ion beam contains the active surface layer. The layer is characterised by carbonised structures consisting of condensed aromatic rings, partially depolymerised underneath the layer with hydroxyl groups and high surface energy. The modified surface layer totally lost the initial polyurethane chemical structure after a fluence of 10^16^ ions/cm^2^. The surface layer thickness is observed to be less than the penetration depth of nitrogen ions with 20 keV energy. The surface is not stable and changes with storage time after the treatment. This system activates the polyurethane surface with free radicals on the edge of graphitic clusters which can covalently graft a protein onto the polyurethane with complete coverage of the polymer surface without precursors. The highest activity of the treated surface to attach protein is expected straight after the treatment. The system can be directly installed in a biomedical laboratory or surgery room to activate the polyurethane biomedical device before usage. Future investigations with animal trials are on the way.

## Figures and Tables

**Figure 1 biomimetics-09-00234-f001:**
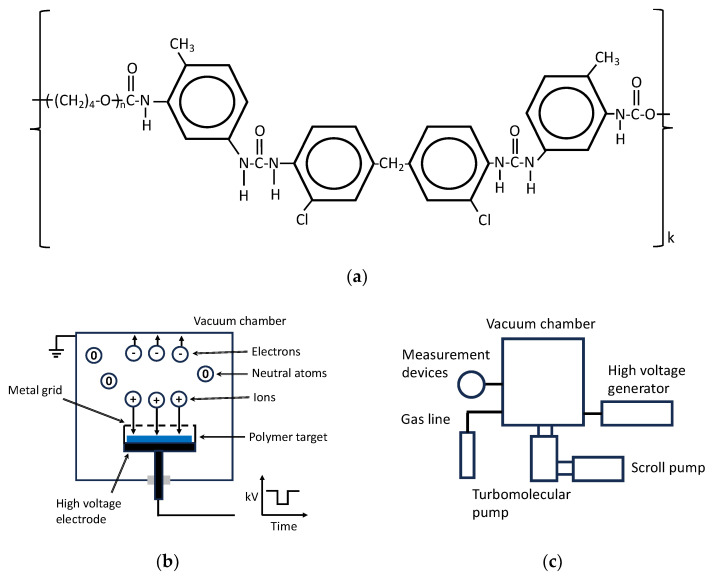
(**a**) The PU formula and (**b**,**c**) scheme of the plasma system. (**b**) The scheme of the plasma chamber with a high voltage electrode, grid, and polymer sample (blue rectangular). The plasma is generated during a short, high-voltage pulse. Ions from plasma are accelerated towards the polymer surface. (**c**) A block scheme of the system, including vacuum pumps, gas line, high voltage generator, and sensors of pressure, voltage, current, and plasma spectrum (marked as measurement devices).

**Figure 2 biomimetics-09-00234-f002:**
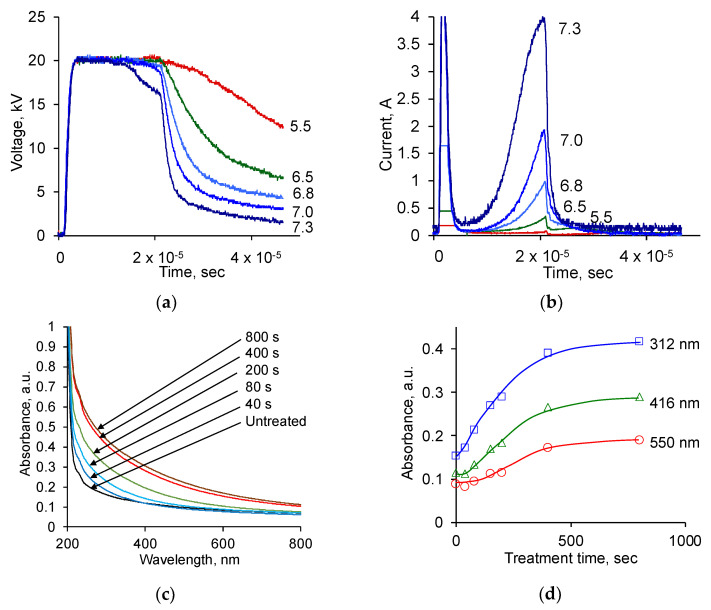
(**a**) Voltage oscillogram during pulses at different pressures (marked in mTorr) in a vacuum chamber; (**b**) current oscillogram during pulses at different pressures (marked in mTorr) in a vacuum chamber; (**c**) UV spectra of PE films treated for various durations (for fluence calculation) (see in text); (**d**) UV spectra absorbance of PE films at different wavelengths to determine the fluence of ion beam treatment (for fluence calculation) (see in text).

**Figure 3 biomimetics-09-00234-f003:**
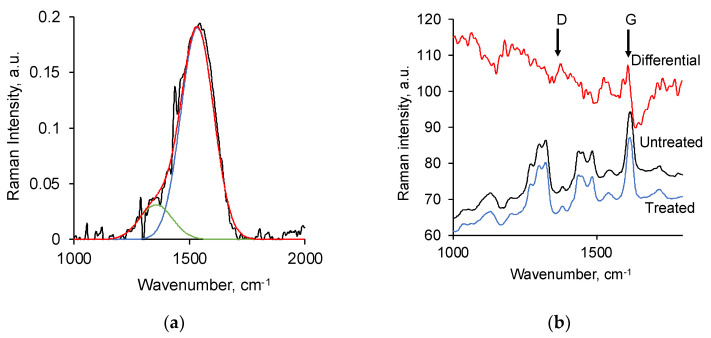
Micro-Raman spectra of (**a**) PE treated by an ion beam, black line is the experimental spectrum, green line is D-peak, blue line is G-peak, red is their sum; (**b**) untreated PU, treated PU by an ion beam and their difference.

**Figure 4 biomimetics-09-00234-f004:**
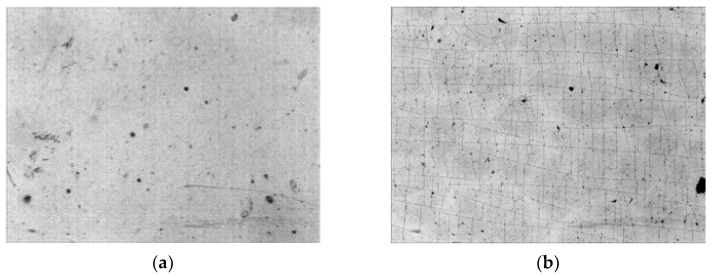
Micro-photo of: (**a**) untreated PU film; (**b**) PU film treated by ion beam. The size of both images is 1.2 mm × 0.9 mm.

**Figure 5 biomimetics-09-00234-f005:**
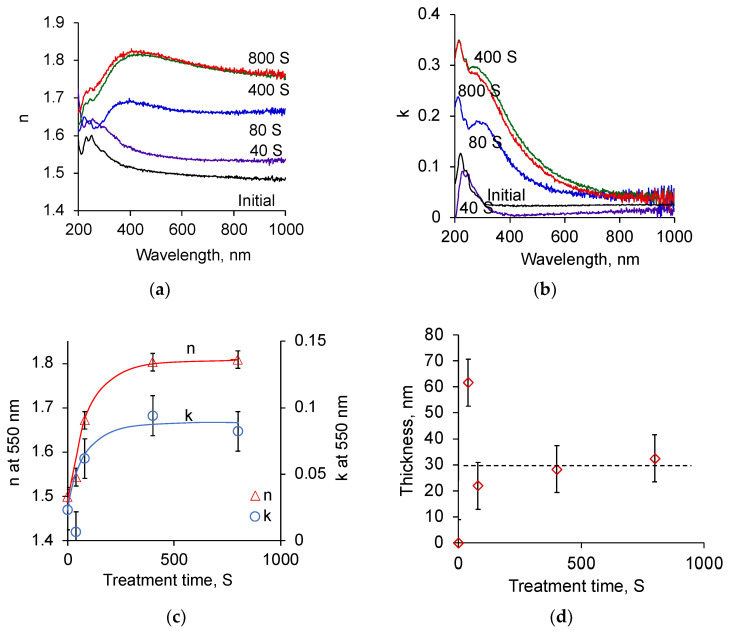
Optical ellipsometry spectra and results of ion beam treated PU: (**a**) spectra of the refractive index of the top layer for different treatment time (marked in s); (**b**) spectra of the extinction coefficient of the top layer for different treatment time (marked in s); (**c**) refractive index and extinction coefficient of the top layer in dependence on treatment time; (**d**) thickness of the top layer in dependence on treatment time.

**Figure 6 biomimetics-09-00234-f006:**
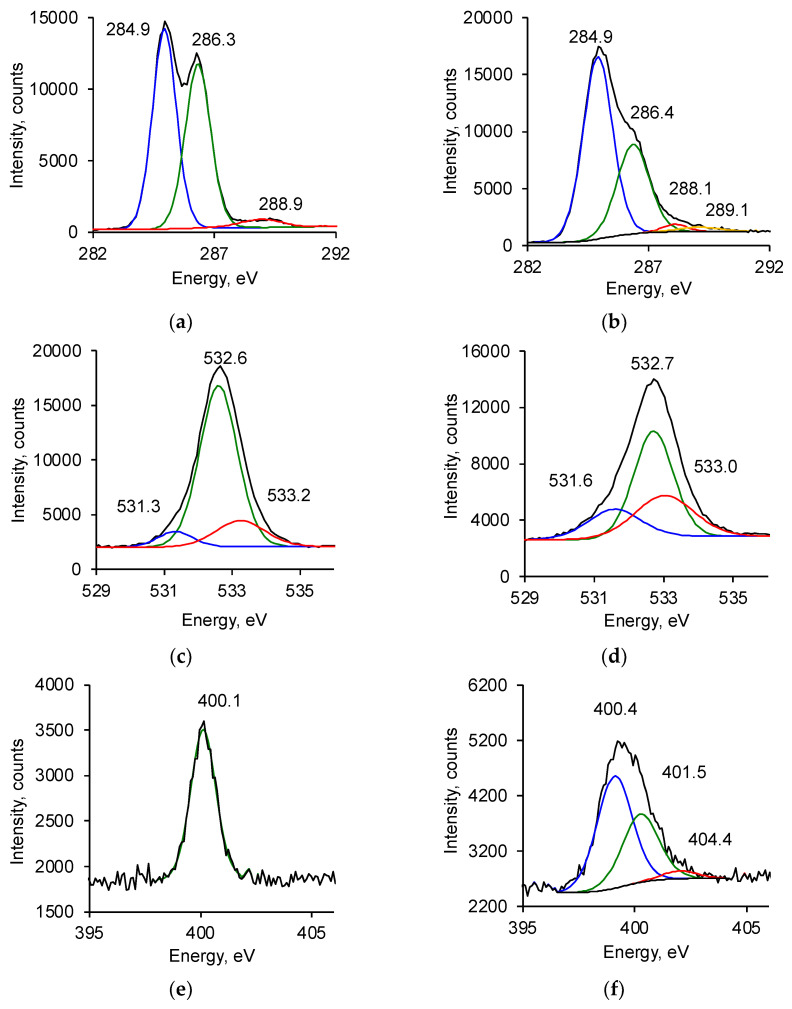
X-ray photoelectron spectra (XPS) of untreated (**a**,**c**,**e**) and treated (**b**,**d**,**f**) PU films: (**a**,**b**) C_1s_ region; (**c**,**d**) O_1s_ region; and (**e**,**f**) N_1s_ region. The experimental lines are fitted with Gauss functions (color lines). Position of maximum is marked in eV.

**Figure 7 biomimetics-09-00234-f007:**
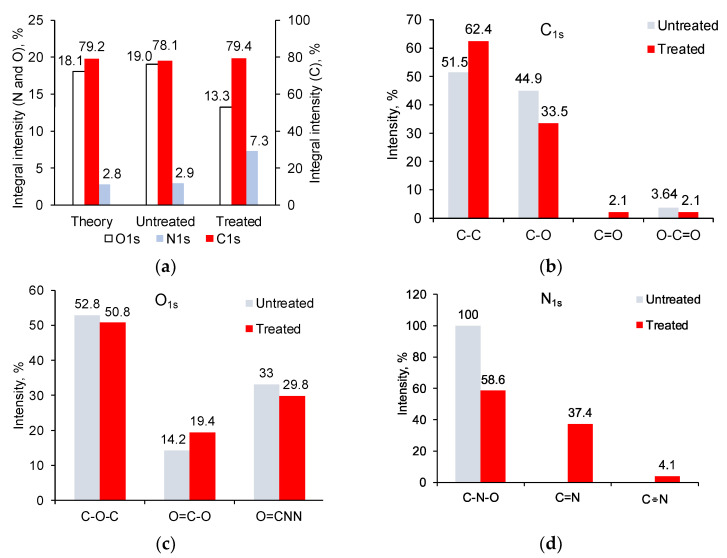
The element content of untreated and treated PU surface provided by XPS data: (**a**) total content; (**b**) content of carbon atoms in different bonds by high-resolution C1s line; (**c**) content of oxygen atoms in different bonds by high-resolution O1s line; and (**d**) content of nitrogen atoms in different bonds by high-resolution N1s line.

**Figure 8 biomimetics-09-00234-f008:**
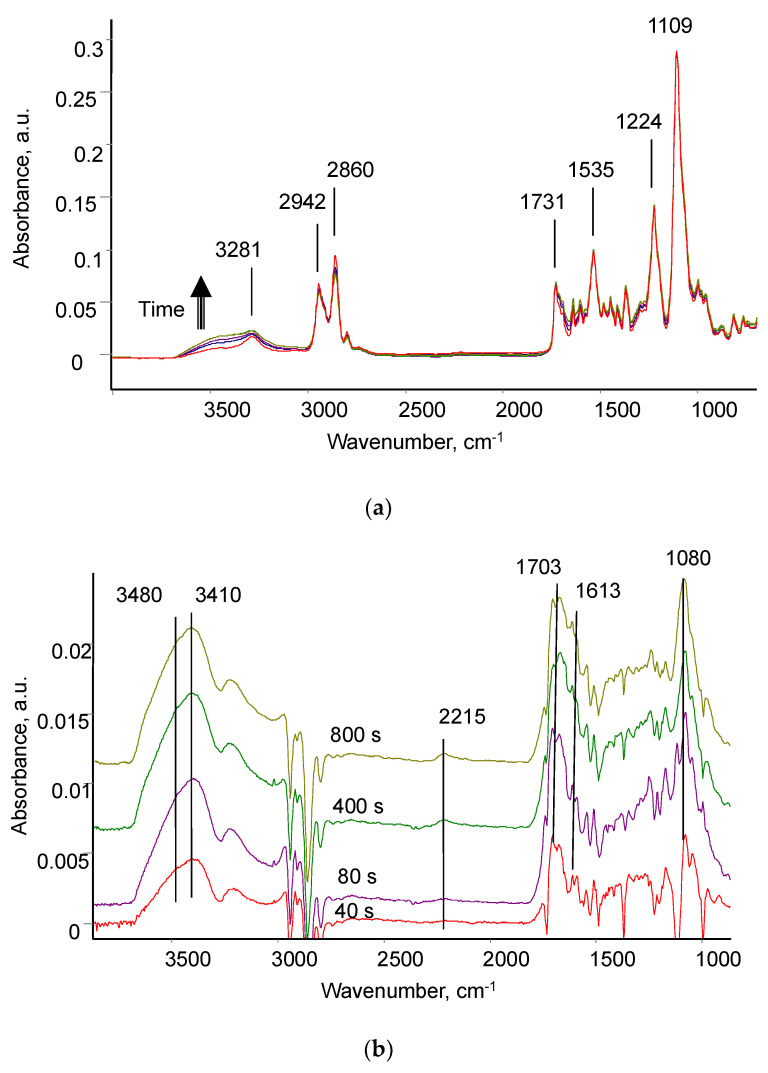
FTIR ATR spectra of PU films treated with treatment different times: (**a**) original spectra, spectral change with treatment time is shown with array; (**b**) differential spectra (spectrum of untreated PU is subtracted). Different color lines correspond to different treatment time.

**Figure 9 biomimetics-09-00234-f009:**
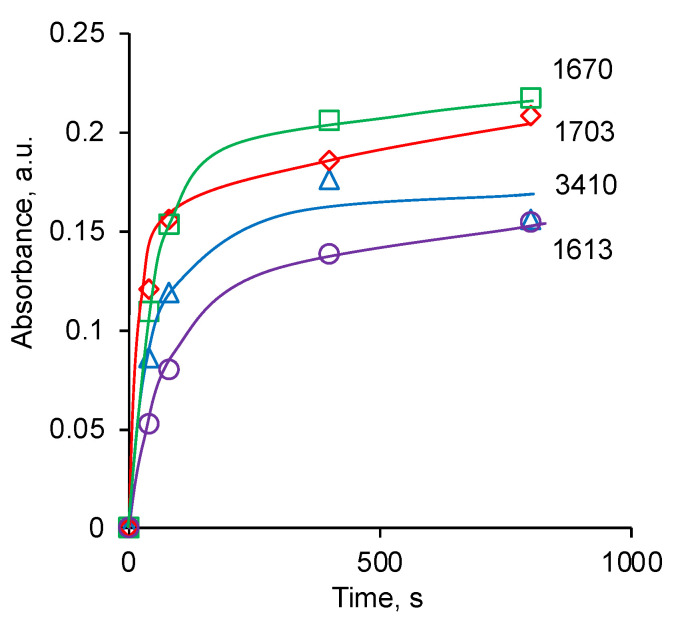
The absorbance of spectral lines in the FTIR ATR spectra of PU depends on treatment time.

**Figure 10 biomimetics-09-00234-f010:**
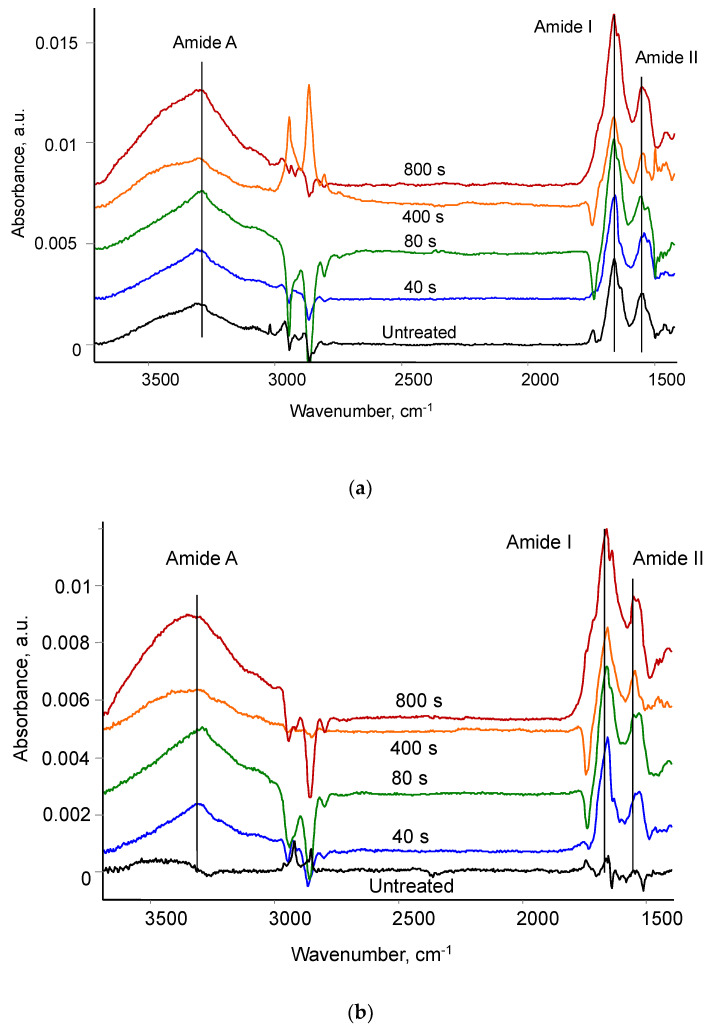
Differential spectra of PU surface with attached fibrinogen: (**a**) fibrinogen is attached and washed in the buffer; (**b**) fibrinogen is attached and washed with detergent. The spectrum of PU is subtracted. The treatment time of polyurethane in an ion beam is shown.

**Figure 11 biomimetics-09-00234-f011:**
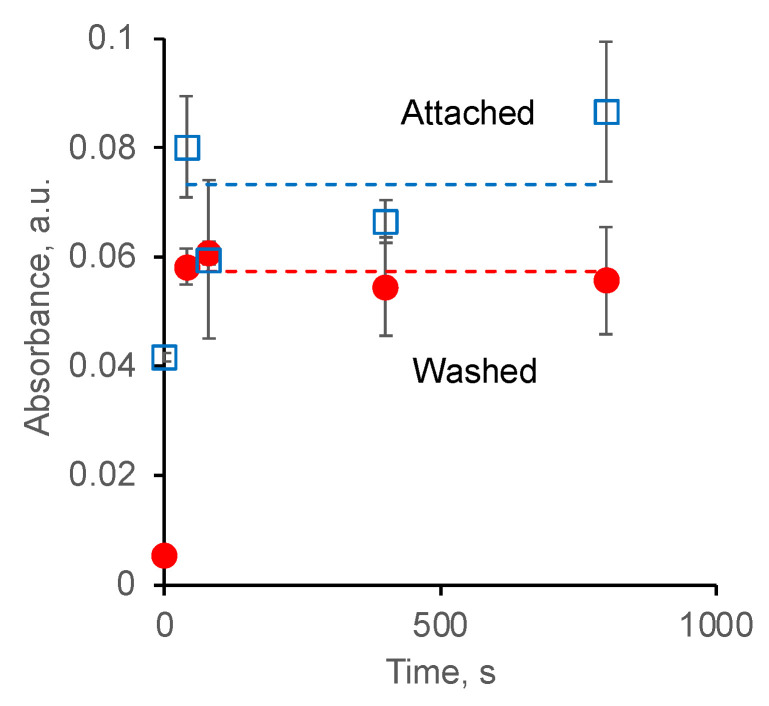
The absorbance of fibrinogen lines in FTIR ATR spectra of PU with attached fibrinogen before and after washing in detergent.

**Table 1 biomimetics-09-00234-t001:** Wetting contact angles (degree) and surface energy (mJ/m^2^) of PU untreated and ion beam treated of 800 s.

Title 1	Title 2		Title 3		
	Water	CH_2_I_2_	Total	Polar	Dispersic
Untreated	94	55	32.6	1.2	31.4
Treated after 30 min	37	39	65.4	25.3	40.1
Treated, after 1 month	57	50	50.6	16.4	34.3

## Data Availability

Data are available on request from the corresponding author.

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
