# Peer review of "Attachment of Fibrinogen on Ion Beam Treated Polyurethane"

_biomimetics, 2024, doi:10.3390/biomimetics9040234_

Round 1

Reviewer 1 Report

Comments and Suggestions for Authors

Dear Authors,

This manuscript describes a method for attaching fibrinogen on ion beam treated polyurethane. Surface and structure analyses were performed to highlight the changes occurred in the surface layer of the treated polyurethane. The resulting surface of polyurethane became capable of adsorbing and chemically binding protein and represents a promising scaffold for protein coverage in biomedical polymer devices.   

The manuscript offers a lot of information regarding the procedure of surface modification using ion beam treatment, with detailed explanations and analyses. The results are quite well correlated.  However, after a careful reading of the manuscript, there are some observations and questions to address:

1)      The introduction is a little bit too long, with too many details. Please try to highlight the main ideas objectively. For example, the description of the method that uses a ion beam can be moved to the experimental part.

2)      Lines 83-84 it appears twice “ Typlically”

3)      Please summarize the last paragraph and better describe the novelty and the advantages of the method.

4)      A schema summarizing the steps of surface modification (like a graphical abstract) would be useful

5)      The paragraph from line 342-349 that describes the changes in wettability of the polyurethane could be better improved by summarizing the data in a table. Measuring the contact angle with liquid of different polarity is useful to calculate the surface energy (Owens-Wendt method). This is one of the parameters that determines the future behavior of the surfaces regarding the protein’s adsorption or in contact with the living tissue.

6)      The Conclusions are very general and should highlight better the complexity, the results of the study and the future perspectives.

Author Response

Answers on referee’s comments

Referee 1.

1) The introduction is a little bit too long, with too many details. Please try to highlight the main ideas objectively. For example, the description of the method that uses a ion beam can be moved to the experimental part.

Answer: The introduction contains a problem discussion, methods of solution and actual task for the present investigation. The length of Introduction is determined by complexity of the problem and existing attempts to solve it. To show clear the task, we have to describe what is done and what is not done. The ion beam techniques are important to get the active surface of polymer. If this description is cut, that the task and novelty would be unclear. If the ion beam techniques is moved to Experimental part, the task in the Introduction will be unclear. Besides that, the Experimental part is for description of existing methods and used equipment. It would be unclear, what known methods and devices were used and what new methods were designed. Therefore, we cannot accept this comment.

2) Lines 83-84 it appears twice “ Typlically”

Answer: It has been corrected.

3) Please summarize the last paragraph and better describe the novelty and the advantages of the method.

Answer: The last paragraph is extended.

4) A schema summarizing the steps of surface modification (like a graphical abstract) would

be useful

Answer: The graphical abstract is added.

5) The paragraph from line 342-349 that describes the changes in wettability of the

polyurethane could be better improved by summarizing the data in a table. Measuring the

contact angle with liquid of different polarity is useful to calculate the surface energy

(Owens-Wendt method). This is one of the parameters that determines the future behavior

of the surfaces regarding the protein’s adsorption or in contact with the living tissue.

Answer: The calculations of the surface energy and its components are added. The data are presented in Table 1 now.

6) The Conclusions are very general and should highlight better the complexity, the results of

the study and the future perspectives.

Answer: The Conclusion is extended.

Reviewer 2 Report

Comments and Suggestions for Authors

see report

Comments on the Quality of English Language

see report

Author Response

  1. The article is so long (and so many – nice – data in there), while the Abstract is so short: whereas it is generally a good quality to keep the Abstract short (a good old rule is <= 200 words), this one is really so short (107 words) that there is room for adding some more details about the work: please specify better the treatment in there (fluence or frequency and duty cycle of pulses, total treatment duration), and main results of physical characterization.

Answer: The Abstract is extended to 183 words. Parameters of the treatment and other details are added.

  1. 123 references are definitely too many for a research article, would be fine for a Review paper: please diminish them at least below one half, ie around 60.

Answer: The references have been revised. Total number is 69 now.

  1. On the first occurrence of word polyurethane, please define acronym PU: while it is used later on in the paper, many times it is not. So, define it on its first occurrence and then use always (meaning: full word should appear only on its first occurrence). The same should be done for polyethylene (PE).

Answer: Acronyms PE and PU as well as others have been corrected.

  1. Introduction: too long. Cutting and/or some parts of it will also help diminish the references.

Answer: This study is based on three points – requirement of strong attachment of protein, ion beam treatment for activation of polymer surface and new plasma system. For all three points we have to clear explain, why we use this way? It means, we have to show the achievements in three different field of science - protein attachment, plasma systems for polymer modification and structural effects in polymers after ion beam treatment. If we cut some field, we could not clarify the novelty of the study.

On the other side, the journal is online only and does not have a limit for an Introduction or length of a manuscript. Therefore, we prefer to keep all information in the Introduction.

  1. “To mimic the surface of an artificial material”, no: an artificial surface is engineered to mimic a natural one, instead. An easy fix is simply replace “mimic” with “modify”.

Answer: It has been corrected following the comment. The “mimic” is replaced with “modify”.

  1. “Such surface preparation methods”: the Authors love the word “such” so much, they use a lot and often misuse it. Here, better: “These surface …”

Again: “Such precursors”: “These p…”

“Such complete coverage”: “This complete…” or “Such a complete…”

“Such implant”: “This implant” or “Such an implant”

“In such systems”, twice: “In these systems”.

“Such negative experiences”: “This negative experience”

“In such case”: “In this case”

“Such plasma sheath”: “This plasma sheath”.

“sizes of such clusters”: “size of these clusters”

Answer: The word “such” has been replaced following the comment.

  1. “without the precursors”: remove completely “the” or replace with “any”.

Answer: The phrase “without the precursors” has been replaced with the phrase “without any precursors” following the comment.

  1. “The covalently attach proteins without precursors to … has been achieved”: 2 mistakes. Fix: “Proteins attached covalently without precursors to… have been achieved”.

Answer: The phrase “The covalently attach proteins without precursors to … has been achieved” has been replaced with phrase “Proteins attached covalently without precursors to… have been achieved”.

  1. “The covalently attaching … has been shown”: fix: “Covalently attached … have been shown”.

Answer: The phrase “The covalently attaching … has been shown” has been replaced with the phrase “Covalently attached … have been shown” following the comment.

  1. “mass separated”: what does it mean?

Answer: The words “mass separated” have been deleted.

  1. “Plasma Immersion Ion Implantation”: do not capitalize initials (same as done for XPS). Same is for “Sodium Dodecyl Sulphate”, “Fourier Transform Infrared”, “Attenuated Total Reflection”.

Answer: It has been corrected.

  1. “To do this”: “To implement this technique”

Answer: This phrase has been changed in two places following the comment.

  1. “the ions are accelerated and, penetrating through the grid, bombard the polymer”: “the ions are accelerated through the grid and bombard the polymer”.

Answer: The sentence has been changed following the comment.

  1. “a size of treated polymer”: “the amount of treated polymer”

Answer: The phrase “a size of treated polymer” has been replaced with “an area of treated polymer”. The “amount of treated polymer” includes also a thickness, when the surface area of polymer is treated independently on thickness.

  1. “by a volume of vacuum chamber”: “by the volume of the vacuum chamber”

Answer: The phrase has been changed following the comments.

  1. “ultimate pressure was up to”: replace “up” with “down”

Answer: The word “up” has been replaced with word “down” following the comments.

  1. “placed on high voltage electrode”: “placed on the cathode”

Answer: The high voltage electrode in this plasma system plays different role including reverse (positive) voltage and floating potential voltage. Therefore, the “cathode” term would be less accurate than more neutral term “high voltage electrode”. Also the term “high voltage electrode” is broadly accepted ion beam and PIII techniques. So, we prefer to use “high voltage electrode” as a more universal term.

  1. “ion fluence was calibrated by UV-spectra of the treated PE film”: unclear. Remove hyphen. Also for “UV-transmission”

Answer: The sentence has been changed to “The ion fluence was calibrated using UV spectra of the treated polyethylene film following the method described in [47].” Hyphens were deleted.

  1. “mg/mL”: I see this is a common habit in biology, writing it like this; but why not simplify to “g/L”?

Answer: It has been replaced with “g/L”.

  1. “pH7”: at pH of 7” or “at pH=7”.

Answer: It has been corrected with “pH = 7”.

  1. “10 minutes”: “10 min”; later on: “2 hours”: “2 h”.

Answer: It has been replaced with “min” and “h”.

  1. “70C”: missing ball

Answer: The ball is added.

  1. “the sample were”: “the samples were”

Answer: It has been corrected.

  1. Cluster size should be given with an error along with it.

Answer: The size of cluster was calculated by formula from Ferrari and Robertson papers. Their theory of cluster size calculation is based on a number of assumptions, which do not allow to estimate an error, because the theory is not enough accurate. This result of cluster size is only for rough estimation. The error of cluster size calculation is based on their assumption, and cannot be based on the half-width of the spectral line or different sample measurements. This is why in all studies the error of the cluster size is not usually presented. It would be incorrect to put the error of the cluster size based on our sample measurements.

  1. Sorry why Fig.s 2c,d and 3a report data of PE not PU?

Answer: Polyethylene films were used for evaluation of the plasma system, comparison with the known results on PIII and ion beam treatment of PE and calculation of ion beam fluence. However, PE has limited biomedical applications. PU biomedical devices are widely used due to high elasticity and softness. Therefore, we evaluate the plasma system with well known results on PE treatment and apply this plasma system on biomedically demanded PU. The detail explanation of this reason is presented in the manuscript.

  1. Conclusions of such a long paper can’t be so void and should be somewhat expanded (same as Abstract).

Answer: The Abstract and Conclusion are extended.

  1. Self-citations are 69-94, 116-118: total 26+3=29. The same for the total, these should be limited to max one half of that.

Answer: The reference list has been shortened following this and other comments. The self-citation number is 6 now. Total number of references is 69. Therefore, the amount of self-citation is 8.7%, that is lower than the widely accepted self-citation in scientific publications (30%) and lower than the number of self-citation in scientific publications accepted in Biomimetics, MDPI (15%).

Round 2

Reviewer 1 Report

Comments and Suggestions for Authors

The corrections have been made and the authors answered the adressed questions. 

However, I did not find in the manuscript the schema summarizing the steps of surface modification, as the authors precised... As the graphical abstract is nor mandatory, I can agree with the manuscript publication in Biomimetic journal.